

# Growth Score: a single metric to define growth in 96-well phenotype assays

Daniel A. Cuevas and Robert A. Edwards

Computational Science Research Center, San Diego State University, San Diego, CA, USA

## ABSTRACT

High-throughput phenotype assays are a cornerstone of systems biology as they allow direct measurements of mutations, genes, strains, or even different genera. High-throughput methods also require data analytic methods that reduce complex time-series data to a single numeric evaluation. Here, we present the Growth Score, an improvement on the previous Growth Level formula. There is strong correlation between Growth Score and Growth Level, but the new Growth Score contains only essential growth curve properties while the formula of the previous Growth Level was convoluted and not easily interpretable. Several programs can be used to estimate the parameters required to calculate the Growth Score metric, including our *PMAnalyzer* pipeline.

# INTRODUCTION

Bacterial growth of homogenous cultures is commonly described through the Monod growth phases (*Monod, 1949*). Specifically, the three major phases of growth are lag phase, where growth rate is zero and bacterial density is constant at the initial measurement; exponential phase, where growth rate is at its maximum value; and stationary phase, where growth rate is zero and bacterial density is constant at its maximum yield. The Growth Level (GL) has been used to quantitatively measure the amount of growth displayed by a bacteria liquid culture (*Cuevas & Edwards, 2017*). In *PMAnalyzer*, an automated growth curve analysis pipeline, *GL* is calculated following the least-squares fitting of the Zwietering logistic model (*Zwietering et al., 1990*):

$$\hat{y} = y_0 + \frac{A - y_0}{1 + \exp\left[\frac{4\mu}{A}(\lambda - t) + 2\right]}.$$

Here, $y_0$ represents the starting absorbance, $\lambda$ represents the lag time, $\mu$ represents the maximum growth rate, $A$ represents the biomass yield obtained during stationary phase, $t$ represents the time vector, and $\hat{y}$ represents the modeled growth curve vector. Using $\hat{y}$, $y_0$, and $A$, Growth Level can be calculated as

$$GL = \frac{n}{\sum_i^n \frac{1}{x_i}},$$

where $x_i = (\hat{y}_i - y_0) + \text{amplitude} = (\hat{y}_i - y_0) + (A - y_0)$.

Corresponding author
Daniel A. Cuevas, dcuevas@sdsu.edu, dcuevas08@gmail.com

*GL* is a variation on the harmonic mean where the logistic growth curve is weighted by the amount of biomass the bacteria culture attained, or the difference in bacterial density, during the course of the experiment (represented as *amplitude* in the *GL* formula). This amplitude-weighted metric performs well for differentiating growing data from growth curves that display no growth. The *GL* provides threshold values that can be used to ascribe qualitative labels or classes to growth, ranging from **no growth** to **very high growth**.

There are several mathematical drawbacks of the *GL* formula. The harmonic mean is commonly used to average values of rates; however, absorbance data are not rate measurements. In addition, using the entire growth curve in *GL* is implicitly affected by the lag time of the bacteria that is dependent on numerous biological properties including ageing of cells, activation of enzymes, metabolic adaptation, and other regulatory mechanisms (*Monod, 1949*; *Robinson et al., 1998*; *Rolfe et al., 2012*). The simplicity of the three phase growth curve is lost in the *GL* calculation.

Here, we propose a new calculation to simplify the quantitative meaning of the level of growth. The Growth Score, *GS*, is defined by three parameters of the Zwietering (*Zwietering et al., 1990*) bacterial growth curve

$$GS = (A - y_0) + 0.25\mu.$$

*GS* uses the starting absorbance, $y_0$, biomass yield, $A$, and the maximum growth rate, $\mu$, to compute a score for a growth curve. Without any explicit dependency on the fitted values or implicit dependencies on lag time, *GS* is clearly understood by its growth parameters. In addition, *GS* performs similarly to *GL*, primarily because of *GL*s strong dependency on yield in defining a quantitative measurement of growth. *GS* has been implemented in the *PMAnalyzer* pipeline (https://edwards.sdsu.edu/pmanalyzer/) and is replacing the *GL* metric.

## SIMULATING GROWTH CURVES

A set of 1,000 growth curves were generated using a total time $t = 240$ h (10 days) and the Zwietering logistic model. Uniformly-distributed random values were selected for each growth curve parameter. Distribution ranges for each parameter were established to portray realistic values: starting absorbance, $y_0 = [0.05, 0.10]$; lag time, $\lambda = [0, 120]$; biomass yield, $A = [0.1, 1.2]$; and maximum growth rate, $\mu = [A/t, 1.1A]$. Growth rate range is limited by the biomass yield to provide a realistic result. The first value represents an organism achieving the biomass yield at the end of the experiment; when $A$ is large this simulates very slow growth. The second value represents a rate where yield is obtained within one hour; when $A$ is large this simulates very fast growth. For cases when $A$ is small, growth rate is negligible, therefore has minimal influence on *GS*.

## COMPARISON OF GROWTH LEVEL AND GROWTH SCORE

Using the simulated growth curves, distributions of *GL* and *GS* are illustrated in the Fig. 1 histograms. The range for *GL* is much larger than *GS*. Both metrics demonstrate a slight left-skewness but *GL* was slightly higher at 0.143 compared to *GS* at 0.078. The growth

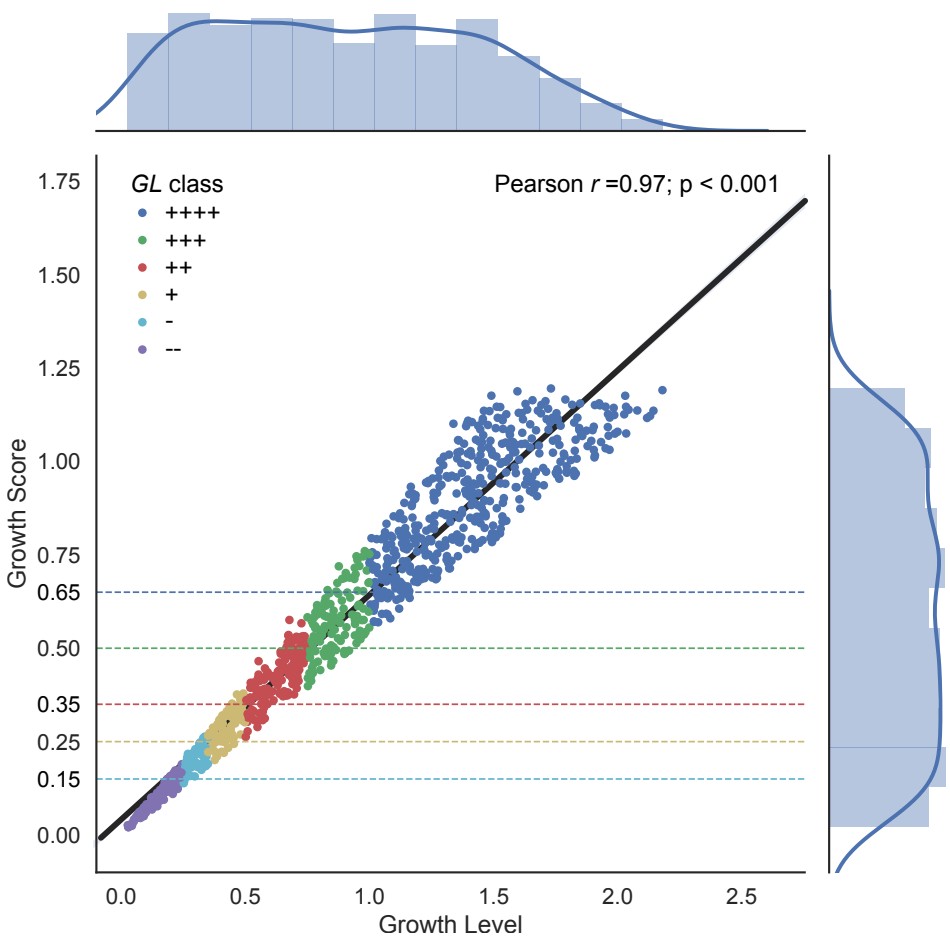

**Figure 1  Growth Level and Growth Score correlation.** The linear relationship results in a Pearson correlation coefficient of 0.97. Distributions are plotted as marginal histograms for each metric. Point colors represent the *GL* classes whereas dotted lines represent the proposed *GS* lower thresholds for each colored class.

score should be left skewed because a low asymptote correlates with low growth rates and results in a low growth score, but the reverse is not necessarily true, high growth score values may result from high or low growth rates.

Differences in metrics are displayed in Fig. 1 with *GL* shown along the *x*-axis and the corresponding *GS* value along the *y*-axis, along with a linear regression line, and growth classes. A strong correlation is shown between *GL* and *GS* with a Pearson correlation coefficient of 0.97 (*p*-value < 0.001) (Fig. 1). The qualitative classes are defined in *PMAnalyzer* as value thresholds. Given these *GL* classes, *GS* class thresholds are also indicated in Fig. 1 as horizontal dotted lines. A total of 191 out of the 1,000 growth curves were classified differently between the two metrics, however, only 20 growth curves had a change from "−" to "+" or vice versa.

To further demonstrate the separation of the growth data, all growth curves are plotted according to their classification in Figs. 2 and 3. In Fig. 2, as growth class decreases from
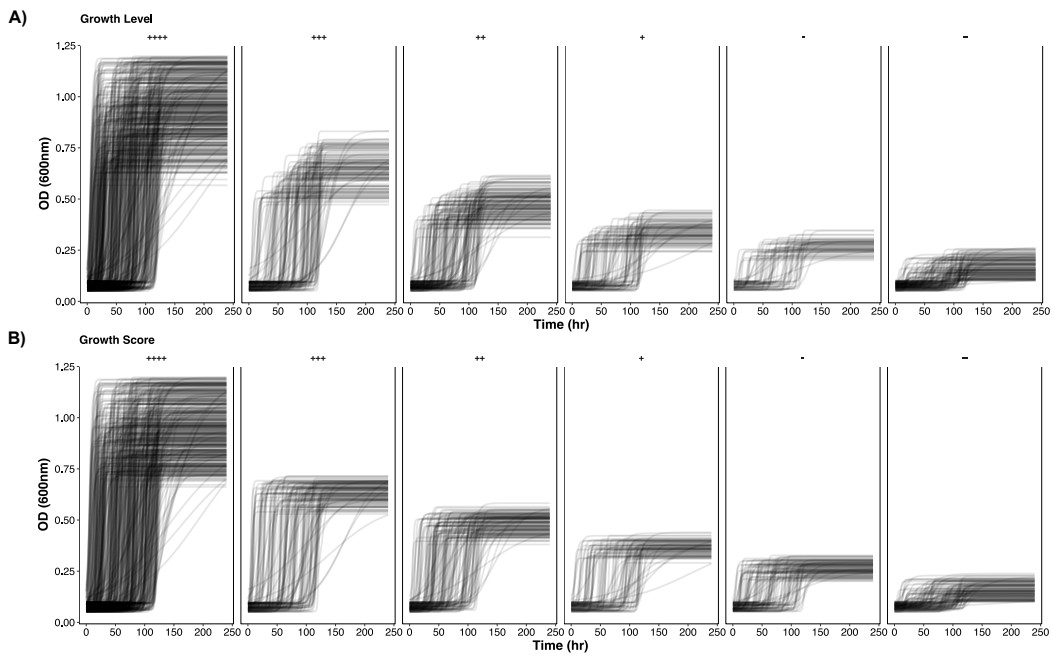

**Figure 2** **Simulated growth curves.** Growth curve data stratified by *GL* and *GS* classes. Classes are organized in decreasing growth from left to right. (A) *GL* classes. (B) *GS* classes.

left to right, the curve height (biomass yield) also decreases. Interestingly while observing biomass yield, *GL* (Fig. 2A) does not separate curves as well as *GS* (Fig. 2B)—there is more overlap in curve height between *GL* classes. Figure 3 comprises of average growth curves per growth class. Standard error intervals are included in graphs but are notably small. In Fig. 3, the "+++" and "++" classes show separation earlier in *GS* (Fig. 3B) than in *GL* (Fig. 3A).

To present more detail, Fig. 4 illustrates distributions between *GL* and *GS* classes per growth curve parameter. The dependence on biomass yield and maximum growth rate is further depicted. Again, biomass yield overlaps between classes much more in *GL* than in *GS*. The widest distribution of yield lies in the highest growth class "++++". This provides opportunity to define more classes of growth to minimize the variation here. Maximum growth rate has less influence in defining higher growth classes, and lag time has no impact on growth class.

Growth curves were generated using the Python 3.6 programming language (Python Language Reference, version 3.6; Python Software Foundation, Wilmington, DE, USA. Available at https://www.python.org), NumPy version 1.13.1 (http://www.numpy.org/) (*Walt, Colbert & Varoquaux, 2011*), pandas version 0.20.3 (http://pandas.pydata.org) (*McKinney, 2010*), and Seaborn version 0.8.1 (http://seaborn.pydata.org/). The ggplot2 library (*Wickham, 2009*, p. 2) in the R programming language (*R Development Core Team, 2008*) was used to generate growth curves in Fig. 2. See Supplemental Information 1 for the Python code that generates the random curves, Supplemental Information 2 for the Jupyter Notebook version, and Supplemental Information 3 for a PDF version of the Jupyter Notebook.
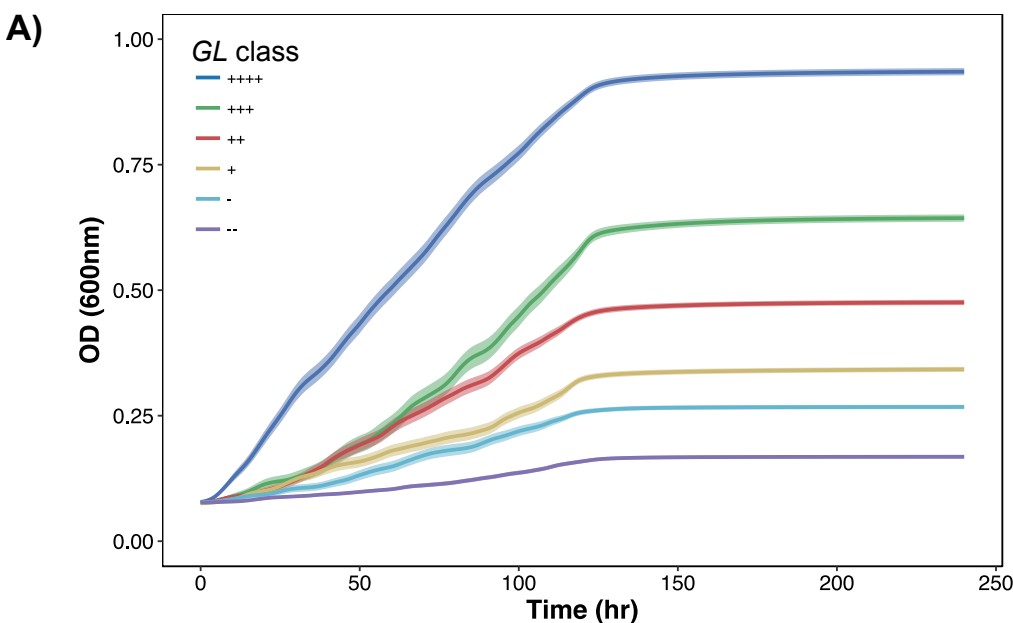

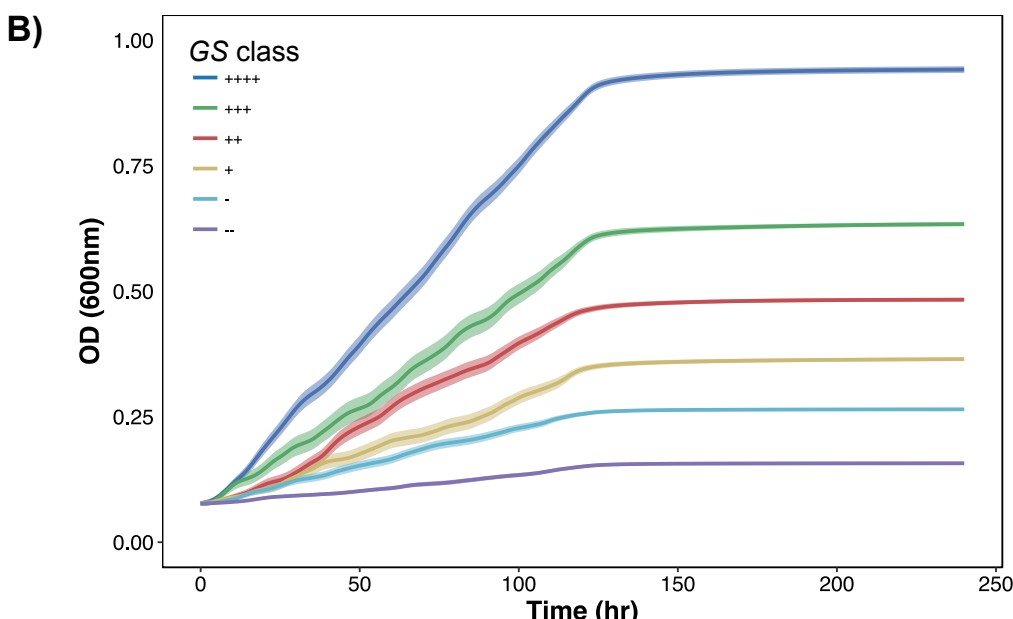

**Figure 3** **Growth curves distributions.** Average growth curves colored by *GL* and *GS* classes. $+/-$ 1 standard error intervals are drawn around the average. (A) *GL* classes. (B) *GS* classes.

Biomass yield has been successful in discriminating growth from no growth using the *GL* formula. However, that calculation lacked direct incorporation of the maximum growth rate, a useful component in measuring fitness. Intuitively, faster growth rates should indicate a bacteria that is more capable of consuming nutrients and proliferating. Yet a

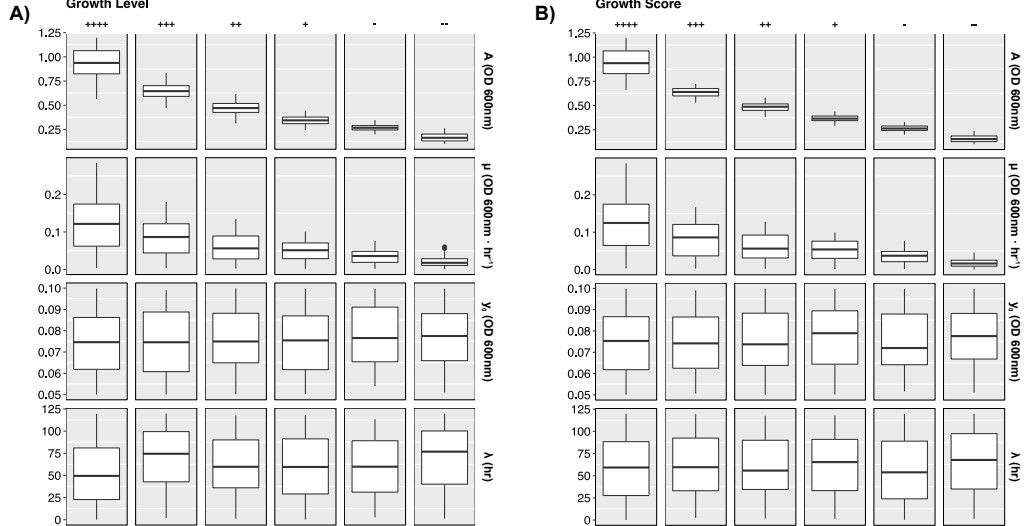

**Figure 4** **Growth curve parameter distributions.** Box plots presenting the median and interquartile ranges for each simulated growth curve parameter. Data are stratified by *GL* and *GS* classes along the columns and by parameters along the rows. Plots demonstrate how each growth metric contains a strong dependence on biomass yield (A) and growth rate (μ). Lag time and starting yield did not have strong effects on growth class. (A) *GL* classes. (B) *GS* classes.

higher rate of growth does not solely demonstrate fitness, evidenced by the abundance of slow-growing bacteria throughout the environment (e.g., *r* and *K* growth strategies (*Pianka, 1970*)). Nor is it a strong feature to ultimately predict larger bacterial densities. For example, two bacteria with the same phenotype can reach the same density, even if one grows at half the rate of the other. Essentially, the ability to reach that potential at a faster rate indicates some biological advantage but has less significance than yield. In *GS* maximum growth rate is included at an amount of 25% in order for biomass accumulation to result in the primary component of growth. Faster growth rates can occur within a short time frame, resulting in low yield and high rate and, therefore, causing a disproportionate *GS* if the magnitude of the growth rate was not reduced.

Here, we have introduced the Growth Score, a new parameter for describing bacterial growth in 96-well phenotypic assays. The Growth Score provides three distinct advantages over other metrics used to describe growth: First, it only uses growth curve properties (yield and growth rate) in its calculation, in contrast our previous Growth Level was averaged over time and was thus heavily influenced by the length of the experiment. Second, Growth Score can be used with results from other software that also performs growth curve modeling or parameterization (DuctApe (*Galardini et al., 2014*), GCAT (*Bukhman et al., 2015*), grofit (*Kahm et al., 2010*; *Vaas et al., 2012*), OmniLog Biolog Phenotype MicroArrays (*Borglin et al., 2012*; *Vaas et al., 2012*) without the need for the raw data, whereas growth level would at least need lag time and /or the raw spectrophotometry data. Finally, the time independence of Growth Score is also a benefit over measurements like Area Under the Curve (AUC) employed by some software. AUC is subject to similar biases as growth

level–longer experiments directly affect how lag time and growth rate mathematically influence the AUC calculation.

### Funding

This work is supported by the National Science Foundation (CNS-1305112 and MCB-1330800). The funders had no role in study design, data collection and analysis, decision to publish, or preparation of the manuscript.

### Grant Disclosures

The following grant information was disclosed by the authors:
National Science Foundation: CNS-1305112, MCB-1330800.

### Competing Interests

The authors declare there are no competing interests.

### Author Contributions

- Daniel A. Cuevas conceived and designed the experiments, performed the experiments, analyzed the data, contributed reagents/materials/analysis tools, prepared figures and/or tables, authored or reviewed drafts of the paper, approved the final draft.
- Robert A. Edwards analyzed the data, authored or reviewed drafts of the paper, approved the final draft.

### Data Availability

    The code to generate the random data set used in the study is provided in a Supplemental File.

### Supplemental Information

Supplemental information for this article can be found online at http://dx.doi.org/10.7717/peerj.4681#supplemental-information.

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
