# Peer review of "Growth Score: a single metric to define growth in 96-well phenotype assays"

_PeerJ, doi:10.7717/peerj.4681_

## Round 0.1 · original submission · Major Revisions

The study merits publication. However, the authors should respond to all comments from the reviewers and address all the suggested changes, especially the comments from reviewer 1 and 2. I agree with reviewer 1 that it is not clear if empirical data (from the literature) was used to validate the new method, if not I suggest to include it.

Reviewer 1 ·

Basic reporting

Cuevas and Edwards have put forward a new method for calculating bacterial growth curves. The study merits publication. My major concern with this study is the lack of details on the simulations and some restrictions of the simulations.

First, authors (line 57) have set some parameters to estimate grwoth curves. Is 50 h enough for the simulation of growth curves? Ok for fast growers, but How about the slow-growing bacteria? Can the authors detail by adding a table with different types of microbes and/or indicating the on the figures?

Second, it is not clear at all the sources of carbon used in this study. The authors mentioned a 96-well phenotype assay, but it is not clear which are these 96 compounds used/or not by bacteria. It requires a table or list.

Third, the study performs a significant number of simulations. It is not clear if empirical data was used to validate the new method. Although figure 1 is a very nice representation of the method, it lacks the details of the dots, the examples of bacterial names.
Line 62->please provide examples of such organisms.

Experimental design

ok.

Validity of the findings

ok.

Additional comments

ok.

Reviewer 2 ·

Basic reporting

Better Mathematical models for fitting bacterial growth are required in microbiology and biotechnology fields in the modern post genome era. Authors develop improved forumula named as the Growth Score, in corvering the defects of lost of three phase growth simplicity of frequently used the Gowth level. The aim of this study is sound, but some minor points are raised in experimentals.

Experimental design

Authors described the GS formula, but biomass yield (A) must be defined clear. Auhtors also mentioned to the GS formula is adopted to 96 well format. However, variety of plate type is now available, it is wondering wheter the GS can adopt to 8 well to 384 well format. The new formula estimating microbial growths can contribute in microbiology and biotechnology fields. Validity and robustness should be tested using in vitro experiments not only in silico.

Validity of the findings

The new formula estimating microbial growths can contribute in microbiology and biotechnology fields. in vitro experiments not only in silico could strengthen the validity and robustness of the GS froumula.

Reviewer 3 ·

Basic reporting

In this manuscript Cuevas and Edwards describe the Growth Score, a new approach for estimating microbial growth based on the analysis of growth curves. This manuscript has clearly stated goals, and is written in a straightforward and clear manner. The methodology and the discussion of the results are sound, and this manuscript deserves to be published. Nevertheless some small issues must be addressed prior to publication.

Experimental design

All the experiments are properly designed and performed.

Validity of the findings

Ln 72-75(Figure 1): Apparently the differences between GS and GL values are not random, and tend to be higher towards higher values of these variables, specially those of the GL class “++++”. Analysing a residuals plot could help to properly quantify this bias, and the authors should at least briefly discuss if it is relevant and why it exists.

Ln 80 (Figure 2 Legend): Not clear to me what exactly is the bootstrapping analysis in this context. Do the authors simply mean that the average curves were calculated based on 100 simulated curves for each category?

Additional comments

Ln 30-31: The meaning of the variables μ and λ should be mentioned here already as it is necessary for the interpretation of the Zwietering logistic model.

Ln 100-111: This paragraph does a great job of clarifying the advantages of the GS metric over GL. Yet I do think it would benefit from having some examples of experiments in which the GS would yield drastically superior results than the GL metric, e.g. when comparing slow versus fast growers, or during metanalysis of growth data was obtained with different experimental times.

---

## Round 0.2 · accepted · Accept

Dear Dr Cuevas,

You have properly enhanced the study according to the reviewers' suggestions. Your paper has been accepted for publication in PeerJ. Congratulations!

# Reviewer 1 ·

Basic reporting

The authors have enhanced their study according to my previous suggestions. New computational simulations have been included, new figures, and discussion.No further comments.

Experimental design

.

Validity of the findings

.

Additional comments

.

Reviewer 3 ·

Basic reporting

The authors have properly addressed all the issues raised in my initial review. I believe this version is suitable for publication.

Experimental design

Experimental design was updated and is adequate.

Validity of the findings

The results support the findings.

Additional comments

Ln 98-106: this paragraph describing the methods feels inadequate for this portion of the manuscript. I recommend that the authors move it to the end.